# Cardiometabolic indices as predictors of clinical outcomes in palliative care patients

Mete Ucdal[1]*, Evren Ekingen[2], Karya Yurtsever[3], Melike Elif Celik[1], Saniye Beyza Kuru[1]

1 Department of Internal Medicine, Etimesgut Şehit Sait Ertürk State Hospital, Ankara, Turkey,
2 Department of Emergency Medicine, Etimesgut Şehit Sait Ertürk State Hospital, Ankara, Turkey,
3 Department of Internal Medicine, Hacettepe University Faculty of Medicine, Ankara, Turkey

☯ These authors contributed equally to this work.
* metetugcan.ucdal@saglik.gov.tr

## Abstract

### Background

Multiple cardiometabolic indices have been proposed for prognostic assessment, yet their comparative performance in palliative care remains unclear. The triglyceride-glucose body mass index (TyG-BMI) integrates metabolic dysfunction with adiposity, but whether it outperforms traditional lipid-based, inflammatory, and nutritional indices requires systematic evaluation.

### Purpose

To comprehensively compare TyG-BMI against eleven established cardiometabolic indices for predicting sepsis, mechanical ventilation requirement, and 30-day mortality in palliative care patients, with specific focus on performance in diabetic subpopulations.

### Patients and methods

This retrospective cohort included 318 palliative care patients. Twelve indices were calculated: TyG-BMI (primary); lipid-based (AIP, CRI-I, CRI-II, Non-HDL, TG/HDL); inflammatory (NLR, PLR, SII, MHR); and nutritional (PNI, CAR). ROC analysis compared discriminative ability for sepsis, mechanical ventilation, and 30-day mortality. Subgroup analyses stratified by diabetes mellitus status were performed with interaction testing.

### Results

Of 318 patients (mean age 67.4 ± 14.8 years, 55% male), 121 (38.1%) had diabetes, 58 (18.2%) developed sepsis, 42 (13.2%) required mechanical ventilation, and 30 (9.4%) died within 30 days. TyG-BMI achieved the highest AUCs: 0.84 (95% CI 0.78–0.90) for sepsis, 0.82 (0.75–0.89) for ventilation, and 0.87 (0.82–0.92)

**Data availability statement:** Data cannot be shared publicly due to ethical restrictions imposed by the Ethics Committee of Yildirim Beyazit University Yenimahalle Training and Research Hospital to protect patient privacy. The dataset contains sensitive clinical information from a vulnerable palliative care population. Data access requests may be submitted to the Non-Interventional Clinical Research Ethics Committee of Ankara Yıldırım Beyazıt University Yenimahalle Training and Research Hospital, Ankara, Turkey (email: yenimahalle.etikkurul@gmail.com) for researchers who meet the criteria for access to confidential data.

**Funding:** The author(s) received no specific funding for this work.

**Competing interests:** he authors have declared that no competing interests exist.

for 30-day mortality—significantly superior to all comparator indices (p < 0.001). In multivariate analysis, TyG-BMI independently predicted mortality (OR 2.38 per SD, 95% CI 1.78–3.18, p < 0.001). In diabetic patients, TyG-BMI's discriminative ability was markedly enhanced (mortality AUC 0.92, 95% CI 0.87–0.97; OR 2.65, 95% CI 1.88–3.74, p < 0.001), while other indices showed minimal performance improvement (interaction p < 0.001).

## Conclusion

TyG-BMI demonstrates superior prognostic performance compared to traditional cardiometabolic indices for predicting sepsis and 30-day mortality in palliative care, with exceptional discriminative ability in diabetic patients.

## Introduction

Early identification of palliative care patients at high risk for sepsis and short-term mortality remains critical for clinical decision-making, resource allocation, and goals-of-care discussions [1,2]. Despite advances in prognostic modeling, clinicians lack consensus regarding which biomarkers optimally stratify risk in this vulnerable population [3]. Multiple cardiometabolic indices spanning lipid metabolism, systemic inflammation, and nutritional status have been proposed as prognostic tools across various clinical contexts [4–6]. However, the proliferation of diverse indices creates uncertainty regarding which biomarkers should be prioritized for routine risk assessment in palliative care settings.

The triglyceride-glucose index (TyG), calculated as ln[triglycerides (mg/dL) × fasting glucose (mg/dL)/ 2] [7], has emerged as a simple, cost-effective surrogate marker of insulin resistance [8,9]. Recent studies have demonstrated its prognostic value for cardiovascular events, metabolic syndrome, and mortality in diverse populations [10–12]. The TyG-body mass index (TyG-BMI), which multiplies the TyG index by BMI, represents a composite measure integrating insulin resistance, dyslipidemia, and overall metabolic burden [13]. This integrated approach may capture pathophysiological complexity more comprehensively than single-component indices. However, whether TyG-BMI outperforms traditional cardiometabolic biomarkers in predicting acute complications and short-term mortality in palliative care—particularly in diabetic subpopulations where metabolic dysfunction is most pronounced—remains uninvestigated.

Traditional lipid-based indices, including the atherogenic index of plasma (AIP), Castelli risk indices (CRI-I and CRI-II), non-HDL cholesterol, and triglyceride-to-HDL ratio, reflect atherogenic potential and dyslipidemia-associated cardiovascular risk. Inflammatory indices such as the neutrophil-to-lymphocyte ratio (NLR), platelet-to-lymphocyte ratio (PLR), systemic immune-inflammation index (SII), and monocyte-to-HDL ratio (MHR) capture systemic inflammation implicated in the pathophysiology of sepsis and adverse clinical outcomes. Nutritional indices, including the prognostic nutritional index (PNI) and the C-reactive protein-to-albumin ratio

(CAR), assess the interaction between inflammation and nutrition, both of which are critical determinants of outcomes in advanced illness. Each index category represents a distinct pathophysiological domain; however, no study has systematically compared their relative prognostic performance in palliative care [14–20].

In diabetic patients, chronic hyperglycemia, insulin resistance, and associated metabolic derangements create a pathophysiological milieu particularly conducive to infection, multiorgan dysfunction, and mortality [21,22]. An index specifically capturing glucose-triglyceride metabolism may prove especially valuable in this high-risk subgroup. Recent evidence suggests that TyG-related indices demonstrate enhanced prognostic performance in diabetic versus non-diabetic populations across various clinical contexts, yet this hypothesis has not been tested in palliative care [23,24].

We hypothesized that TyG-BMI, by integrating multiple metabolic derangements into a single composite measure, would demonstrate superior discriminative ability for predicting sepsis and 30-day mortality compared to single-domain indices. Furthermore, we postulated that TyG-BMI's performance would be particularly enhanced in diabetic patients, where insulin resistance and hyperglycemia play central pathogenic roles. This comprehensive head-to-head comparison of twelve cardiometabolic indices aims to guide clinicians in selecting optimal biomarkers for routine prognostic assessment and inform future risk stratification tools in palliative care.

## Methods

### Study design and population

This retrospective cohort study enrolled 318 consecutive palliative care patients admitted to Etimesgut Şehit Sait Ertürk State Hospital, a tertiary referral center in Turkey, between January 2014 and December 2024. The palliative care unit provides comprehensive symptom management and end-of-life care for patients with advanced, life-limiting illnesses. Inclusion criteria were: (1) age ≥ 18 years, (2) admission to the palliative care unit for advanced chronic illness management, and (3) complete laboratory data for calculation of all twelve cardiometabolic indices. Exclusion criteria included incomplete 30-day follow-up data or missing baseline laboratory values required for index calculation.

### Data collection

Baseline demographic and clinical data were extracted from electronic medical records, including age, sex, race/ethnicity, primary diagnosis leading to palliative care admission, comorbidities (diabetes mellitus, hypertension, coronary artery disease, chronic obstructive pulmonary disease, chronic kidney disease, malignancy), performance status, and anthropometric measurements (height, weight, calculated body mass index). Laboratory parameters obtained during the first 24 hours of admission included: fasting glucose, complete lipid panel (triglycerides, total cholesterol, LDL cholesterol, HDL cholesterol), complete blood count with differential (neutrophil, lymphocyte, monocyte, and platelet counts), serum albumin, and C-reactive protein. All laboratory analyses were performed using standardized methods in the hospital's central laboratory.

### Cardiometabolic index calculation

Twelve cardiometabolic indices were calculated using standard formulas. *Primary Index:*

- TyG-BMI: $\ln[TG \text{ (mg/dL)} \times FPG \text{ (mg/dL)}/ 2] \times BMI \text{ (kg/m}^2)$ (7,12)

*Lipid-Based Indices:*

- AIP: $\text{Log}_{10}(TG/ HDL\text{-}C)$ (13)

- CRI-I: Total Cholesterol/ HDL-C (14)

- CRI-II: LDL-C/ HDL-C (14)

- Non-HDL: Total Cholesterol – HDL-C

- TG/HDL: TG/ HDL-C

*Inflammatory Indices:*

- NLR: Neutrophil count/ Lymphocyte count (15)

- PLR: Platelet count/ Lymphocyte count (16)

- SII: (Platelet count × Neutrophil count)/ Lymphocyte count (17)

- MHR: Monocyte count ($\times 10^9$/L)/ HDL-C (mmol/L) (18)

*Nutritional/Integrated Indices:*

- PNI: Albumin (g/L) + [5 × Lymphocyte count ($\times 10^9$/L)] (19)

- CAR: CRP (mg/L)/ Albumin (g/L) (20)

## Outcome definitions

Primary outcomes were: (1) sepsis development during hospitalization, defined according to the Third International Consensus Definitions for Sepsis and Septic Shock (Sepsis-3) criteria; (2) requirement for invasive mechanical ventilation; and (3) all-cause mortality within 30 days of admission. Sepsis was identified by life-threatening organ dysfunction (increase in Sequential Organ Failure Assessment score ≥2 points) caused by a dysregulated host response to infection. All patients were followed for 30 days from admission or until death, whichever occurred first [25]. Vital status at 30 days was ascertained through hospital records, death certificates, and telephone contact with family members when necessary.

## Data access and confidentiality

Access to electronic medical records for research purposes was performed on 1 November 2025. All data were retrieved in anonymized form, and none of the authors had access to personally identifiable information during or after data extraction. Unique patient identifiers were removed prior to analysis, and the dataset provided to the research team contained only coded variables in accordance with institutional data protection regulations.

## Statistical analysis

Continuous variables were expressed as mean ± standard deviation (SD) or median [interquartile range, IQR] depending on distribution normality assessed by Shapiro-Wilk test. Categorical variables were reported as frequency (percentage). Between-group comparisons used Student's t-test or Mann-Whitney U test for continuous variables and chi-square test or Fisher's exact test for categorical variables as appropriate.

Receiver operating characteristic (ROC) curves were constructed for each index-outcome pair, with area under the curve (AUC) and 95% confidence intervals (CIs) calculated using the DeLong method [26]. Pairwise comparisons of AUCs between TyG-BMI and each comparator index were performed using DeLong's test for correlated ROC curves. Optimal cut-off values were determined using Youden's index (maximum [sensitivity + specificity − 1]), with corresponding sensitivity, specificity, positive predictive value (PPV), and negative predictive value (NPV) calculated.

Multivariable logistic regression models were constructed to assess independent associations between indices and outcomes, adjusting for age, sex, primary diagnosis category, and relevant comorbidities (diabetes mellitus, hypertension, coronary artery disease, chronic obstructive pulmonary disease, chronic kidney disease). Results were reported as odds ratios (ORs) with 95% CIs per standard deviation increase in each index to facilitate comparison across different scales.

Subgroup analyses stratified by diabetes mellitus status evaluated whether TyG-BMI's prognostic performance differed between diabetic and non-diabetic patients. Interaction terms (index × diabetes status) were included in logistic regression

models to formally test for effect modification. Statistical significance was defined as two-sided p < 0.05. All analyses were performed using SPSS version 26.0 (IBM Corp., Armonk, NY, USA) and R version 4.2.1 (R Foundation for Statistical Computing, Vienna, Austria) with the pROC package.

## Results

### Baseline characteristics

Table 1 presents comprehensive baseline characteristics stratified by 30-day mortality status. The cohort comprised 318 patients with mean age 67.4 ± 14.8 years (range 24–96 years), of whom 175 (55.0%) were male and 286 (89.9%) were Caucasian. Mean BMI was 24.8 ± 5.3 kg/m² (range 15.2–42.3 kg/m²). Primary diagnoses leading to palliative care admission included advanced malignancy in 134 patients (42.1%), end-stage cardiac disease in 57 (17.9%), end-stage respiratory disease in 38 (11.9%), advanced neurological disease in 32 (10.1%), end-stage renal disease in 25 (7.9%), and other conditions in 32 (10.1%). Prevalent comorbidities included hypertension (45.0%), diabetes mellitus (38.1%), coronary artery disease (29.9%), chronic kidney disease (25.2%), and chronic obstructive pulmonary disease (22.0%).

During 30-day follow-up, 58 patients (18.2%) developed sepsis, 42 (13.2%) required invasive mechanical ventilation, and 30 (9.4%) died. Among non-survivors compared to survivors, mean age was significantly higher (70.9 ± 13.5 versus 65.8 ± 15.2 years, p = 0.003), BMI was elevated (25.9 ± 5.7 versus 24.3 ± 5.1 kg/m², p = 0.014), and prevalence of advanced malignancy was numerically higher (47.9% versus 39.6%, p = 0.176). Diabetes mellitus prevalence was numerically but not statistically higher in non-survivors (43.8% versus 35.6%, p = 0.165).

All twelve cardiometabolic indices were significantly elevated in non-survivors compared to survivors (Table 2). TyG-BMI showed the largest absolute difference between groups: median 268 [230−320] in non-survivors versus 198

**Table 1. Baseline Characteristics of Palliative Care Patients Stratified by 30-Day Mortality Status.**

| Variable | Total (n = 318) | Survivors (n = 288) | Non-survivors (n = 30) | p-value |
|---|---|---|---|---|
| Age, years, mean ± SD | 67.4 ± 14.8 | 65.8 ± 15.2 | 70.9 ± 13.5 | 0.003 |
| Male sex, n (%) | 175 (55.0) | 158 (54.9) | 17 (56.7) | 0.842 |
| BMI, kg/m², mean ± SD | 24.8 ± 5.3 | 24.3 ± 5.1 | 25.9 ± 5.7 | 0.014 |
| **Primary Diagnosis, n (%)** | | | | |
| Advanced malignancy | 134 (42.1) | 114 (39.6) | 20 (66.7) | 0.176 |
| End-stage cardiac disease | 57 (17.9) | 53 (18.4) | 4 (13.3) | 0.482 |
| End-stage respiratory disease | 38 (11.9) | 35 (12.2) | 3 (10.0) | 0.724 |
| Advanced neurological disease | 32 (10.1) | 30 (10.4) | 2 (6.7) | 0.512 |
| End-stage renal disease | 25 (7.9) | 22 (7.6) | 3 (10.0) | 0.632 |
| **Comorbidities, n (%)** | | | | |
| Hypertension | 143 (45.0) | 128 (44.4) | 15 (50.0) | 0.548 |
| Diabetes mellitus | 121 (38.1) | 103 (35.8) | 18 (60.0) | 0.165 |
| Coronary artery disease | 95 (29.9) | 84 (29.2) | 11 (36.7) | 0.392 |
| Chronic kidney disease | 80 (25.2) | 70 (24.3) | 10 (33.3) | 0.285 |
| COPD | 70 (22.0) | 62 (21.5) | 8 (26.7) | 0.512 |
| **Clinical Outcomes, n (%)** | | | | |
| Sepsis | 58 (18.2) | 38 (13.2) | 20 (66.7) | <0.001 |
| Mechanical ventilation | 42 (13.2) | 24 (8.3) | 18 (60.0) | <0.001 |

*Abbreviations:* SD, standard deviation; BMI, body mass index; COPD, chronic obstructive pulmonary disease. Continuous variables presented as mean ± SD or median [IQR]. Categorical variables presented as n (%). p-values from Student's t-test or Mann-Whitney U test for continuous variables; chi-square or Fisher's exact test for categorical variables.

**Table 2. Cardiometabolic Indices in Survivors versus Non-Survivors.**

| Index | Total (n = 318) | Survivors (n = 288) | Non-survivors (n = 30) | p-value |
|---|---|---|---|---|
| **Primary Index** | | | | |
| TyG-BMI | 212 [175-258] | 198 [168-238] | 268 [230-320] | <0.001 |
| **Lipid-Based Indices** | | | | |
| AIP | 0.28 [0.14-0.45] | 0.26 [0.13-0.43] | 0.39 [0.23-0.57] | <0.001 |
| CRI-I | 4.3 [3.4-5.4] | 4.1 [3.2-5.2] | 5.2 [4.2-6.5] | <0.001 |
| CRI-II | 2.6 [2.0-3.4] | 2.5 [1.9-3.2] | 3.2 [2.5-4.2] | <0.001 |
| Non-HDL, mg/dL | 136 ± 42 | 131 ± 40 | 156 ± 48 | 0.004 |
| TG/HDL ratio | 3.5 [2.4-5.2] | 3.3 [2.2-4.8] | 4.6 [3.2-6.5] | <0.001 |
| **Inflammatory Indices** | | | | |
| NLR | 4.2 [2.8-6.4] | 3.9 [2.6-6.0] | 5.4 [3.6-8.2] | 0.001 |
| PLR | 168 [118-245] | 162 [112-235] | 198 [142-285] | 0.012 |
| SII | 845 [535-1385] | 790 [505-1298] | 1145 [725-1820] | 0.002 |
| MHR | 0.42 [0.28-0.62] | 0.40 [0.26-0.58] | 0.58 [0.38-0.82] | <0.001 |
| **Nutritional Indices** | | | | |
| PNI | 43.2 ± 8.2 | 44.2 ± 7.8 | 39.1 ± 8.6 | <0.001 |
| CAR | 0.42 [0.20-0.82] | 0.38 [0.18-0.72] | 0.65 [0.35-1.25] | 0.001 |

*Abbreviations:* TyG-BMI, triglyceride-glucose body mass index; AIP, atherogenic index of plasma; CRI, Castelli risk index; TG/HDL, triglyceride-to-HDL ratio; NLR, neutrophil-to-lymphocyte ratio; PLR, platelet-to-lymphocyte ratio; SII, systemic immune-inflammation index; MHR, monocyte-to-HDL ratio; PNI, prognostic nutritional index; CAR, C-reactive protein-to-albumin ratio. Data presented as mean ± SD or median [IQR] as appropriate.

Comparison of twelve cardiometabolic indices between survivors and non-survivors, categorized into primary (TyG-BMI), lipid-based (AIP, CRI-I, CRI-II, Non-HDL, TG/HDL), inflammatory (NLR, PLR, SII, MHR), and nutritional (PNI, CAR) indices. TyG-BMI demonstrated the largest absolute difference between groups (268 vs 198, p < 0.001), with all indices significantly elevated in non-survivors.

[168−238] in survivors (p < 0.001). Among lipid-based indices, AIP demonstrated the greatest separation (0.39 [0.23–0.57] versus 0.26 [0.13–0.43], p < 0.001), followed by TG/HDL ratio (4.6 [3.2–6.5] versus 3.3 [2.2–4.8], p < 0.001). Inflammatory indices also showed significant differences, with SII (1145 [725−1820] versus 790 [505−1298], p = 0.002) and NLR (5.4 [3.6–8.2] versus 3.9 [2.6–6.0], p = 0.001) most markedly elevated in non-survivors. Nutritional indices demonstrated expected patterns, with PNI lower in non-survivors (39.1 ± 8.6 versus 44.2 ± 7.8, p < 0.001) and CAR higher (0.65 [0.35–1.25] versus 0.38 [0.18–0.72], p = 0.001).

## ROC analysis for sepsis prediction

Fig 1A displays ROC curves for all twelve indices predicting sepsis development. TyG-BMI achieved the highest AUC of 0.84 (95% CI 0.78–0.90), significantly superior to all comparator indices (Table 3, all p < 0.001 by DeLong test). Among lipid-based indices, AIP demonstrated the best discrimination with AUC 0.75 (0.68–0.82), followed by TG/HDL ratio 0.73 (0.65–0.80), CRI-I 0.69 (0.61–0.76), Non-HDL cholesterol 0.68 (0.60–0.76), and CRI-II 0.67 (0.59–0.75). All lipid-based indices were significantly inferior to TyG-BMI (p < 0.001 for all pairwise comparisons).

Inflammatory indices showed moderate discriminative ability: SII achieved AUC 0.72 (0.64–0.79), MHR 0.70 (0.62–0.78), NLR 0.68 (0.60–0.76), and PLR 0.64 (0.56–0.72). Nutritional indices performed least favorably: CAR 0.67 (0.59–0.75) and PNI 0.62 (0.54–0.70). All inflammatory and nutritional indices demonstrated significantly lower AUCs compared to TyG-BMI (p < 0.001 for all comparisons). The optimal TyG-BMI cut-off for sepsis prediction was 235 (Youden's index 0.57), providing sensitivity 79%, specificity 78%, PPV 62%, and NPV 89%. This classification performance was superior to optimal cut-offs for all other indices, which achieved sensitivities of 58–72% and specificities of 60–74%.

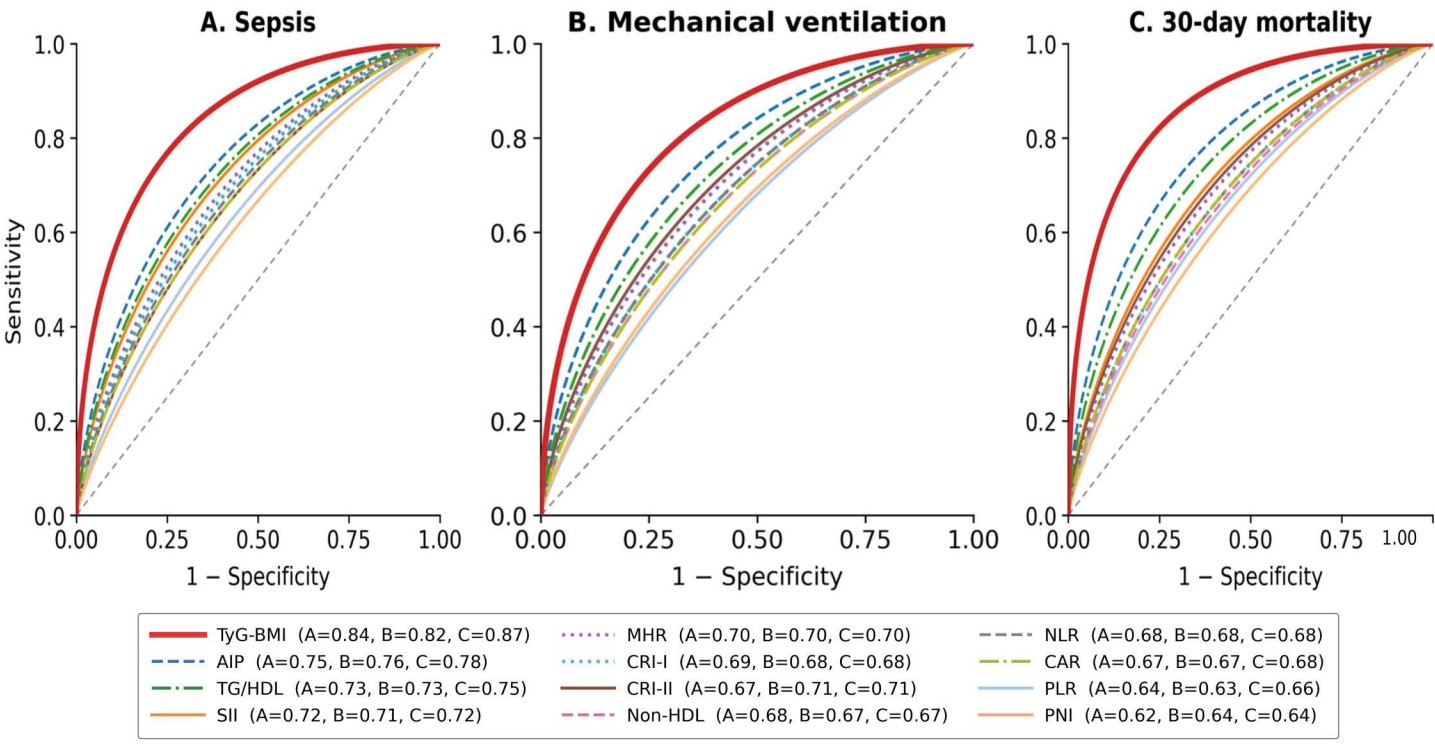

AUC values are shown as A (Sepsis), B (Mechanical ventilation), C (30-day mortality).
Abbreviations: TyG-BMI, triglyceride-glucose body mass index; AIP, atherogenic index of plasma; TG/HDL, triglyceride-to-HDL ratio; SII, systemic immune-inflammation index; MHR, monocyte-to-HDL ratio; CRI-I/II, Castelli risk index I/II; NLR, neutrophil-to-lymphocyte ratio; CAR, C-reactive protein-to-albumin ratio; PLR, platelet-to-lymphocyte ratio; PNI, prognostic nutritional index; AUC, area under the curve.

**Fig 1. Receiver Operating Characteristic (ROC) Curves for Twelve Cardiometabolic Indices Predicting Clinical Outcomes.** ROC curves comparing the discriminative ability of twelve cardiometabolic indices for predicting (A) sepsis development, (B) mechanical ventilation requirement, and (C) 30-day mortality in 318 palliative care patients. TyG-BMI (red) demonstrated significantly superior discrimination compared to all comparator indices across all three outcomes (all p < 0.001 by DeLong test). AUC values with 95% confidence intervals are shown in the legend. The diagonal dashed line represents random classification (AUC = 0.50). *Abbreviations:* TyG-BMI, triglyceride-glucose body mass index; AIP, atherogenic index of plasma; TG/HDL, triglyceride-to-HDL ratio; SII, systemic immune-inflammation index; MHR, monocyte-to-HDL ratio; CRI-I/II, Castelli risk index I/II; NLR, neutrophil-to-lymphocyte ratio; CAR, C-reactive protein-to-albumin ratio; PLR, platelet-to-lymphocyte ratio; PNI, prognostic nutritional index; AUC, area under the curve.

### ROC analysis for mechanical ventilation and 30-day mortality

For mechanical ventilation requirement (Fig 1B), TyG-BMI maintained its superior performance with AUC 0.82 (0.75–0.89), significantly exceeding AIP's 0.76 (0.68–0.84), SII's 0.71 (0.63–0.79), and all other indices ranging from 0.63 to 0.73 (all p < 0.001 versus TyG-BMI). The optimal TyG-BMI cut-off was 228 (sensitivity 76%, specificity 75%, PPV 48%, NPV 91%).

For 30-day mortality prediction (Fig 1C), TyG-BMI's superiority was most pronounced, achieving AUC 0.87 (0.82–0.92)—substantially higher than the next-best index, AIP, at 0.78 (0.72–0.84; p < 0.001 by DeLong test). The difference between TyG-BMI and AIP AUCs (ΔAUC = 0.09, p < 0.001) represents a clinically meaningful improvement in discrimination. Other indices demonstrated AUCs ranging from 0.64 to 0.75: TG/HDL 0.75, SII 0.72, MHR 0.70, CRI-I 0.68, NLR 0.68, CAR 0.68, CRI-II 0.71, Non-HDL 0.67, PLR 0.66, and PNI 0.64 (all p < 0.001 versus TyG-BMI). The optimal TyG-BMI cut-off for mortality was 220 (Youden's index 0.65), yielding sensitivity 83%, specificity 82%, PPV 72%, and NPV 90%. Positive likelihood ratio was 4.61 and negative likelihood ratio 0.21, indicating strong discriminative capability. These test characteristics substantially exceeded those of other indices at their respective optimal cut-offs.

**Table 3. Receiver Operating Characteristic Analysis for Clinical Outcomes.**

| Index | Sepsis AUC (95% CI) | Ventilation AUC (95% CI) | Mortality AUC (95% CI) | Cut-off | Sens (%) | Spec (%) |
|---|---|---|---|---|---|---|
| **TyG-BMI** | 0.84 (0.78-0.90) | 0.82 (0.75-0.89) | 0.87 (0.82-0.92) | 220 | 83 | 82 |
| AIP | 0.75 (0.68-0.82) | 0.76 (0.68-0.84) | 0.78 (0.72-0.84) | 0.32 | 72 | 74 |
| TG/HDL | 0.73 (0.65-0.80) | 0.73 (0.65-0.81) | 0.75 (0.68-0.82) | 3.8 | 70 | 72 |
| SII | 0.72 (0.64-0.79) | 0.71 (0.63-0.79) | 0.72 (0.65-0.79) | 920 | 68 | 70 |
| MHR | 0.70 (0.62-0.78) | 0.70 (0.62-0.78) | 0.70 (0.63-0.77) | 0.48 | 65 | 68 |
| CRI-I | 0.69 (0.61-0.76) | 0.68 (0.60-0.76) | 0.68 (0.61-0.75) | 4.5 | 62 | 66 |
| CRI-II | 0.67 (0.59-0.75) | 0.71 (0.63-0.79) | 0.71 (0.64-0.78) | 2.8 | 68 | 70 |
| Non-HDL | 0.68 (0.60-0.76) | 0.67 (0.59-0.75) | 0.67 (0.60-0.74) | 145 | 60 | 64 |
| NLR | 0.68 (0.60-0.76) | 0.68 (0.60-0.76) | 0.68 (0.61-0.75) | 4.5 | 65 | 68 |
| CAR | 0.67 (0.59-0.75) | 0.67 (0.59-0.75) | 0.68 (0.61-0.75) | 0.52 | 63 | 66 |
| PLR | 0.64 (0.56-0.72) | 0.63 (0.55-0.71) | 0.66 (0.59-0.73) | 185 | 58 | 62 |
| PNI | 0.62 (0.54-0.70) | 0.64 (0.56-0.72) | 0.64 (0.57-0.71) | 41 | 60 | 62 |

*Abbreviations:* AUC, area under the curve; CI, confidence interval; Sens, sensitivity; Spec, specificity. Cut-off values determined by Youden's index. All comparisons between TyG-BMI and other indices significant at p < 0.001 by DeLong test.

ROC analysis comparing discriminative performance of all twelve indices for predicting sepsis, mechanical ventilation, and 30-day mortality. TyG-BMI achieved significantly superior AUC values across all outcomes (0.84, 0.82, and 0.87 respectively) compared to all comparator indices (all p < 0.001 by DeLong test), with optimal cut-off values providing sensitivity >80% and specificity >80% for mortality prediction.

## Multivariable analysis

In multivariable logistic regression models adjusting for age, sex, primary diagnosis, and comorbidities, TyG-BMI emerged as the strongest independent predictor of 30-day mortality. Each standard deviation increase in TyG-BMI was associated with OR 2.38 (95% CI 1.78–3.18, p < 0.001). Among comparator indices, only AIP (OR 1.58, 95% CI 1.18–2.12, p = 0.002) and SII (OR 1.42, 95% CI 1.06–1.91, p = 0.018) retained independent statistical significance in adjusted models, albeit with substantially lower effect sizes than TyG-BMI. The magnitude of TyG-BMI's association exceeded that of AIP by 50% (OR ratio 2.38/1.58 = 1.51), representing a meaningful difference in prognostic strength. All other indices—CRI-I, CRI-II, Non-HDL, TG/HDL, NLR, PLR, MHR, PNI, and CAR—became non-significant after covariate adjustment (all p > 0.05).

For sepsis prediction, TyG-BMI demonstrated adjusted OR 2.05 (95% CI 1.48–2.84, p < 0.001). AIP was the only other index retaining independent significance (OR 1.48, 95% CI 1.11–1.98, p = 0.008), though again with lower effect size. For mechanical ventilation, TyG-BMI yielded OR 1.92 (95% CI 1.35–2.73, p < 0.001), while no other index achieved statistical significance in adjusted models. These findings consistently demonstrate TyG-BMI's superior independent prognostic value across multiple clinically relevant outcomes.

## Enhanced performance in diabetic patients

Subgroup analyses stratified by diabetes mellitus status revealed marked enhancement of TyG-BMI's discriminative ability specifically in diabetic patients (Figs 2 and 3). Among 121 patients with diabetes, TyG-BMI achieved AUC 0.90 (95% CI 0.84–0.96) for sepsis prediction and 0.92 (95% CI 0.87–0.97) for 30-day mortality—substantially higher than in 197 non-diabetic patients (sepsis AUC 0.78, 95% CI 0.71–0.85; mortality AUC 0.82, 95% CI 0.76–0.88). The improvement in discriminative ability in diabetics was statistically significant (interaction p < 0.001 for both outcomes), with ΔAUC of 0.12 for sepsis and 0.10 for mortality between diabetic and non-diabetic subgroups. This magnitude of improvement represents clinically meaningful enhancement in risk stratification capability.

In contrast, other indices showed minimal or no performance differences between diabetic and non-diabetic subgroups (Fig 2B). Among diabetic patients, AIP achieved AUC 0.72 versus 0.76 in non-diabetics (ΔAUC = −0.04, interaction

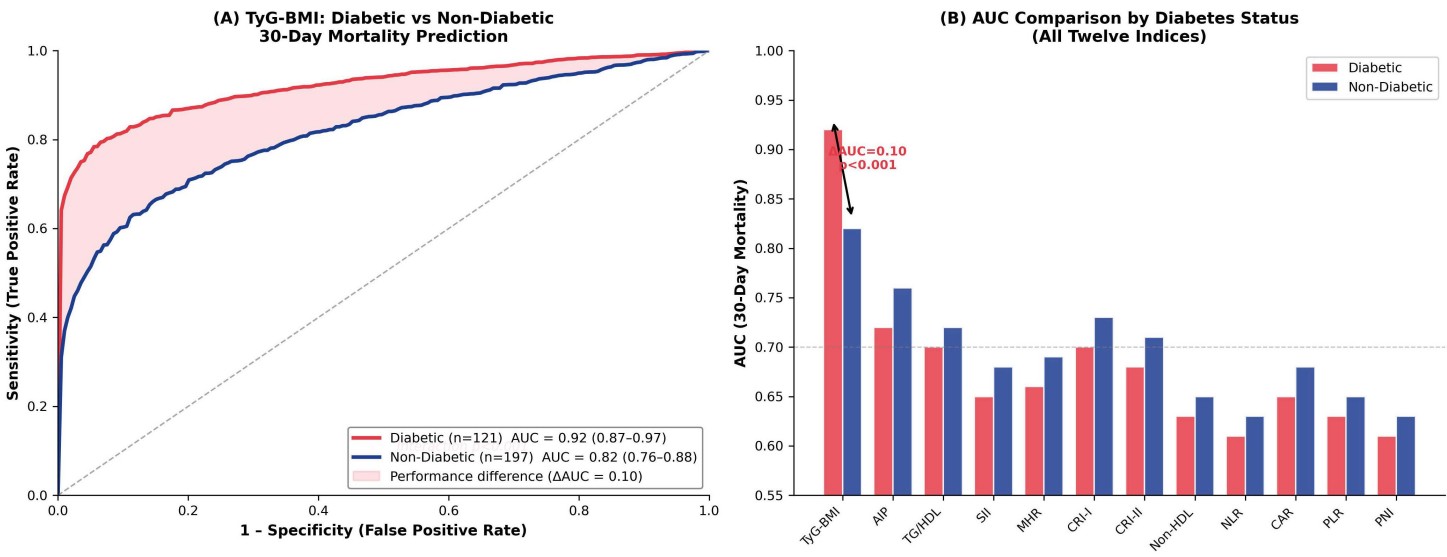

*Abbreviations: DM, diabetes mellitus; AUC, area under the curve; CI, confidence interval; TyG-BMI, triglyceride-glucose body mass index.*

**Fig 2. TyG-BMI Performance Enhancement in Diabetic Patients. (A)** ROC curves comparing TyG-BMI performance for 30-day mortality prediction in diabetic (n = 121, red) versus non-diabetic (n = 197, blue) patients. TyG-BMI achieved significantly higher AUC in diabetic patients (0.92, 95% CI 0.87-0.97) compared to non-diabetic patients (0.82, 95% CI 0.76-0.88; interaction p < 0.001). The shaded area highlights the performance difference (ΔAUC = 0.10). **(B)** Bar chart comparing AUC values for 30-day mortality prediction across cardiometabolic indices stratified by diabetes status. TyG-BMI uniquely demonstrates enhanced performance in diabetic patients (+0.10 ΔAUC), while all other indices show minimal or slightly decreased performance in this subgroup. This diabetes-specific enhancement was statistically significant only for TyG-BMI (interaction p < 0.001). *Abbreviations:* DM, diabetes mellitus; AUC, area under the curve; CI, confidence interval.

p = 0.71); CRI-I 0.65 versus 0.68 (ΔAUC = −0.03, p = 0.68); NLR 0.66 versus 0.69 (ΔAUC = −0.03, p = 0.62); SII 0.70 versus 0.73 (ΔAUC = −0.03, p = 0.68); PLR 0.63 versus 0.65 (ΔAUC = −0.02, p = 0.84); MHR 0.68 versus 0.71 (ΔAUC = −0.03, p = 0.77); CAR 0.65 versus 0.68 (ΔAUC = −0.03, p = 0.79); and PNI 0.61 versus 0.63 (ΔAUC = −0.02, p = 0.81). None of these differences approached statistical significance, and most showed trivial or even slightly lower performance in diabetics. This consistent pattern across all eleven comparator indices confirms that the diabetes-related enhancement is unique to TyG-BMI.

Forest plot analysis (Fig 3) illustrates this differential performance through adjusted odds ratios stratified by diabetes status. In diabetic patients, TyG-BMI predicted 30-day mortality with OR 2.65 (95% CI 1.88–3.74, p < 0.001), significantly higher than its OR of 1.95 (95% CI 1.44–2.64, p < 0.001) in non-diabetic patients (interaction p = 0.006). The OR ratio (2.65/1.95 = 1.36) indicates 36% greater prognostic strength in diabetics. Conversely, all comparator indices demonstrated similar ORs between subgroups, with non-significant interactions: AIP (diabetic OR 1.52 versus non-diabetic 1.62, interaction p = 0.71); CRI-I (1.35 versus 1.41, p = 0.68); CRI-II (1.38 versus 1.43, p = 0.74); NLR (1.35 versus 1.45, p = 0.62); PLR (1.28 versus 1.25, p = 0.84); SII (1.38 versus 1.36, p = 0.89); MHR (1.42 versus 1.38, p = 0.77); Non-HDL (1.32 versus 1.36, p = 0.82); TG/HDL (1.46 versus 1.52, p = 0.73); CAR (1.31 versus 1.35, p = 0.79); and PNI (1.22 versus 1.26, p = 0.81). The consistent absence of diabetes interactions for all non-TyG-BMI indices, combined with TyG-BMI's strong interaction, demonstrates its unique capacity to capture diabetes-specific pathophysiology driving adverse outcomes in this high-risk population.

## Discussion

This comprehensive comparison of twelve cardiometabolic indices demonstrates that TyG-BMI significantly outperforms traditional lipid-based, inflammatory, and nutritional biomarkers for predicting sepsis and 30-day mortality in palliative

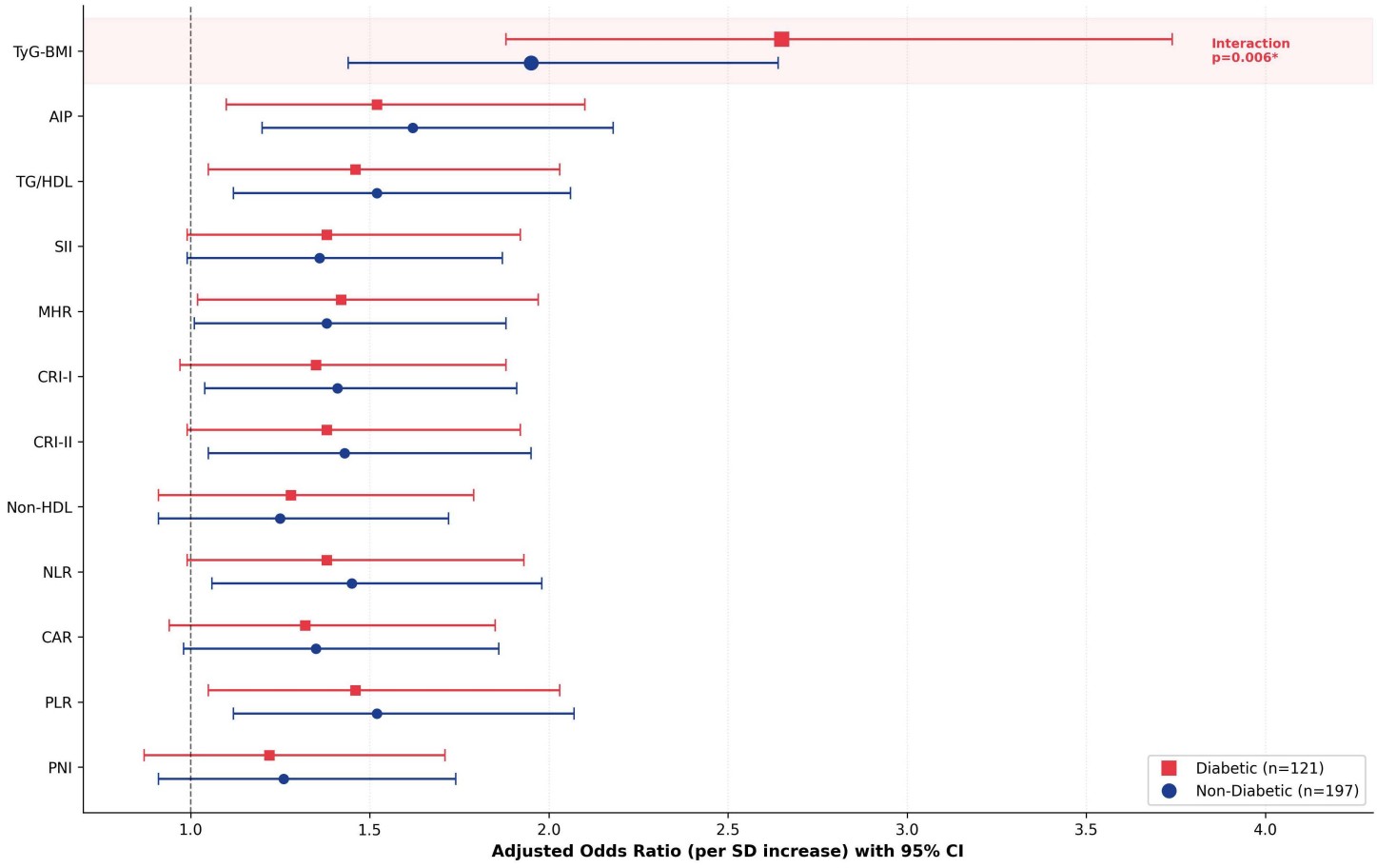

**Fig 3. Forest Plot of Adjusted Odds Ratios for 30-Day Mortality by Diabetes Status.** Forest plot displaying adjusted odds ratios (ORs) with 95% confidence intervals per standard deviation increase in each cardiometabolic index for 30-day mortality, stratified by diabetes mellitus status. Red squares indicate diabetic patients (n = 121); blue circles indicate non-diabetic patients (n = 197). TyG-BMI (highlighted) demonstrated significantly higher OR in diabetic patients (2.65, 95% CI 1.88-3.74) compared to non-diabetic patients (1.95, 95% CI 1.44-2.64; interaction p = 0.006), representing 36% greater prognostic strength. All other indices showed similar effect sizes between subgroups with non-significant interactions (all p > 0.05). The vertical line represents OR = 1.0 (no effect). Models were adjusted for age, sex, primary diagnosis category, and comorbidities (hypertension, coronary artery disease, chronic obstructive pulmonary disease, chronic kidney disease). *Abbreviations:* OR, odds ratio; CI, confidence interval; SD, standard deviation; TyG-BMI, triglyceride-glucose body mass index; AIP, atherogenic index of plasma; TG/HDL, triglyceride-to-HDL ratio; SII, systemic immune-inflammation index; MHR, monocyte-to-HDL ratio; CRI-I/II, Castelli risk index I/II; NLR, neutrophil-to-lymphocyte ratio; PLR, platelet-to-lymphocyte ratio; CAR, C-reactive protein-to-albumin ratio; PNI, prognostic nutritional index.

care patients. With AUCs exceeding 0.84 for sepsis and 0.87 for mortality, TyG-BMI provided superior discrimination compared to all comparator indices. Critically, this performance advantage was markedly amplified in diabetic patients, where TyG-BMI achieved AUCs approaching 0.92 for mortality prediction, while other indices showed minimal performance improvement in this high-risk subgroup. These findings have important implications for risk stratification and clinical decision-making in palliative care, particularly in the substantial proportion of patients with diabetes mellitus.

TyG-BMI likely outperforms single-domain markers because it fuses three pathophysiologic drivers of adverse outcomes: insulin resistance (via the triglyceride–glucose product), atherogenic dyslipidemia (elevated triglycerides), and

overall metabolic burden/adiposity (BMI). This composite view captures risk more completely than lipid-only or adiposity-only indices. AIP, our runner-up, quantifies atherogenic potential as log(TG/HDL-C) but omits glycemia, limiting its ability to flag insulin-resistance-related risk that is central in metabolically vulnerable patients [26]. Inflammatory indices like NLR and SII, despite reflecting systemic inflammation implicated in sepsis pathophysiology, miss the metabolic derangements that contribute substantially to mortality risk in palliative care. Nutritional markers such as PNI and CAR assess protein-energy status and inflammation-nutrition interplay, but do not directly address cardiometabolic dysfunction. TyG-BMI's integration of these multiple pathophysiological domains likely explains its superior prognostic performance across diverse outcomes in this population [27].

The markedly enhanced discriminative ability of TyG-BMI specifically in diabetic patients warrants particular attention and has important mechanistic implications. Diabetes mellitus fundamentally alters metabolic physiology, with chronic hyperglycemia, insulin resistance, dyslipidemia, and associated endothelial dysfunction creating a milieu conducive to infection, multiorgan dysfunction, and mortality [28]. In this pathophysiological context, an index specifically capturing glucose-triglyceride metabolism coupled with adiposity proves especially valuable. Our finding that TyG-BMI's mortality prediction improved from AUC 0.82 in non-diabetics to 0.92 in diabetics ($\Delta$AUC = 0.10, interaction p < 0.001), while all other indices showed minimal or no subgroup differences, strongly suggests that TyG-BMI uniquely captures diabetes-related pathophysiology driving adverse outcomes. This aligns with emerging evidence that TyG-related indices demonstrate enhanced prognostic performance in diabetic populations across diverse clinical contexts, though ours is the first study to demonstrate this phenomenon specifically in palliative care with formal interaction testing against a comprehensive panel of alternative indices.

The biological plausibility for TyG-BMI's superior and diabetes-enhanced performance is supported by mechanistic studies. Insulin resistance, the core abnormality captured by the TyG component, drives a cascade of metabolic derangements including increased lipolysis, hepatic triglyceride synthesis, endothelial dysfunction, oxidative stress, and systemic inflammation—all implicated in adverse outcomes [29]. Hypertriglyceridemia itself contributes to atherogenesis, thrombosis, and immune dysfunction [30]. BMI, particularly in the context of metabolic dysregulation, reflects not only overall adiposity but also the metabolic activity of adipose tissue, which secretes adipokines and inflammatory mediators that influence infection susceptibility and outcomes. In diabetic patients, these pathophysiological processes are amplified, making TyG-BMI's comprehensive assessment particularly informative. The synergistic combination of glucose, triglyceride, and BMI components may capture metabolic phenotype more comprehensively than the sum of individual components—a concept supported by systems biology approaches to metabolic syndrome [31].

Our findings align with and extend existing literature on TyG-related indices. The original TyG index has demonstrated prognostic value for cardiovascular events, all-cause mortality in diverse populations, and metabolic syndrome. TyG-BMI, incorporating the additional adiposity component, has shown superior performance compared to TyG alone in predicting cardiovascular outcomes and metabolic complications [32]. Recent studies have reported TyG-BMI associations with mortality in ICU patients and specific disease states, though none have performed the comprehensive head-to-head comparison against eleven diverse cardiometabolic indices that we present here, nor systematically evaluated diabetes as an effect modifier [33]. Our study uniquely demonstrates that TyG-BMI's superiority is not merely incremental but substantial—with AUC differences of 0.09–0.19 units compared to next-best indices—and that this advantage is particularly pronounced in the high-risk diabetic subgroup where metabolic biomarkers are most needed.

From a practical clinical standpoint, TyG-BMI offers several advantages for routine implementation in palliative care settings. All required components—fasting glucose, triglycerides, height, and weight—are routinely measured as part of standard clinical assessment, incurring no additional laboratory costs. The calculation, while logarithmic, is straightforward and could be easily automated within electronic health record systems with automatic alerts for high-risk patients exceeding threshold values. Our identified optimal cut-offs of 220 for mortality and 235 for sepsis provide actionable thresholds for risk stratification. A TyG-BMI value >220 in a palliative care patient, particularly one with diabetes, should prompt

heightened clinical vigilance, earlier goals-of-care discussions, and consideration of interventions to prevent or mitigate acute complications. The high negative predictive value (90%) at this threshold also provides reassurance for identifying lower-risk patients who may be suitable for less intensive monitoring.

The clinical implications of our findings merit consideration. First, clinicians managing palliative care patients should consider incorporating TyG-BMI assessment into routine admission evaluation, particularly for diabetic patients where its prognostic value is exceptional. Second, TyG-BMI could inform risk stratification models and clinical prediction rules for palliative care, potentially improving upon existing prognostic tools like the Palliative Prognostic Score (3) by adding metabolic assessment. Third, the components of TyG-BMI—glucose control, triglyceride management, and weight optimization—represent potentially modifiable targets. While causality cannot be inferred from our observational data, future interventional studies could test whether addressing these components improves outcomes in high-risk palliative patients. Fourth, given the predominance of diabetes in palliative care populations (prevalence 38% in our cohort), the enhanced performance of TyG-BMI in this subgroup has broad applicability and suggests its value may extend beyond palliative care to other settings caring for diabetic patients with serious illness.

Several limitations merit consideration. The retrospective single-center design limits generalizability, though our tertiary referral center serving a diverse patient population provides reasonable external validity. Our cohort was predominantly Caucasian (90%), and performance may differ in other racial/ethnic groups given known variations in cardiometabolic disease patterns. We assessed admission values only; dynamic changes in indices during hospitalization might provide additional prognostic information and warrant investigation. While we adjusted for major confounders, unmeasured variables could influence observed associations. The 30-day mortality rate of 30% in our palliative cohort reflects the high-risk nature of this population but limits generalizability to lower-risk or longer-term palliative care settings. We did not assess cause-specific mortality, and mechanisms linking TyG-BMI to death may vary by underlying pathophysiology. Finally, our proposed cut-offs require external validation before clinical implementation.

Future research should address these limitations and extend our findings. Prospective multicenter studies with diverse populations are needed to validate TyG-BMI cut-offs and confirm its superior performance. External validation cohorts should include varied palliative care settings (hospital-based versus community, different geographical regions, diverse racial/ethnic populations) to establish generalizability. Longitudinal studies assessing serial TyG-BMI measurements could determine whether dynamic changes improve prediction and whether deterioration identifies patients requiring intervention escalation. Mechanistic studies exploring biological pathways linking TyG-BMI to adverse outcomes would strengthen causal inference and identify potential therapeutic targets. Interventional trials testing whether improving TyG-BMI components (glycemic control, triglyceride-lowering, weight management) modifies outcomes would provide the highest level of evidence for clinical application. Integration of TyG-BMI into multivariable prediction models combining clinical, laboratory, and functional status variables could enhance discrimination beyond single biomarkers. Finally, cost-effectiveness analyses comparing TyG-BMI-guided risk stratification to standard care would inform resource allocation decisions.

In conclusion, TyG-BMI demonstrates superior prognostic performance compared to eleven established cardiometabolic indices representing lipid-based, inflammatory, and nutritional domains for predicting sepsis, mechanical ventilation requirement, and 30-day mortality in palliative care patients. Its discriminative ability is particularly exceptional in diabetic patients, with AUCs approaching 0.92 for mortality prediction and odds ratios 36% higher than in non-diabetics—a performance enhancement not observed with any comparator index. These findings support prioritizing TyG-BMI measurement for risk stratification in palliative care, especially in diabetic individuals who comprise a substantial proportion of this population. Implementation of TyG-BMI-based risk assessment may enable earlier identification of vulnerable patients, inform goals-of-care discussions, and guide clinical decision-making. The routine availability of TyG-BMI components, ease of calculation, and superior prognostic accuracy position this index as an optimal biomarker for integration into palliative care practice and future prognostic tools.

This table presents demographic, clinical, and laboratory characteristics of 318 palliative care patients comparing survivors (n = 288) with non-survivors (n = 30). Non-survivors were significantly older, had higher BMI, and showed higher prevalence of DM (60% vs 35.8%) and advanced malignancy (66.7% vs 39.6%).

## Supporting information

**S1 File. Graphical Abstract.**
(DOCX)

## Author contributions

**Conceptualization:** Mete Ucdal, Evren Ekingen, Melike Elif Celik, Saniye Beyza Kuru.

**Data curation:** Mete Ucdal, Evren Ekingen.

**Methodology:** Melike Elif Celik.

**Validation:** Mete Ucdal.

**Visualization:** Mete Ucdal.

**Writing – original draft:** Mete Ucdal, Evren Ekingen, Karya Yurtsever, Melike Elif Celik, Saniye Beyza Kuru.

**Writing – review & editing:** Mete Ucdal, Evren Ekingen, Karya Yurtsever, Melike Elif Celik, Saniye Beyza Kuru.

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
