## [Decision Letter · Decision Letter 0]

25 Jan 2026

PONE-D-25-62734Cardiometabolic Indices as Predictors of Clinical Outcomes in Palliative Care PatientsPLOS One

Dear Dr. ucdal,

Thank you for submitting your manuscript to PLOS ONE. After careful consideration, we feel that it has merit but does not fully meet PLOS ONE’s publication criteria as it currently stands. Therefore, we invite you to submit a revised version of the manuscript that addresses the points raised during the review process.

We look forward to receiving your revised manuscript.

Kind regards,

Ehsan Amini-Salehi

Academic Editor

PLOS One

Journal Requirements:

Additional Editor Comments:

Thank you for submitting your manuscript. After careful review, the manuscript has been evaluated as requiring Major Revisions before it can be considered for publication.

Reviewers' comments:

Reviewer's Responses to Questions

Comments to the Author

1. Is the manuscript technically sound, and do the data support the conclusions?

Reviewer #1: Yes

Reviewer #2: Yes

2. Has the statistical analysis been performed appropriately and rigorously? 

Reviewer #1: Yes

Reviewer #2: Yes

3. Have the authors made all data underlying the findings in their manuscript fully available?

Reviewer #1: No

Reviewer #2: Yes

4. Is the manuscript presented in an intelligible fashion and written in standard English?

Reviewer #1: Yes

Reviewer #2: Yes

5. Review Comments to the Author

Reviewer #1: Dear Authors,

Thank you for targeting this novel idea. To my knowledge, this is the first study evaluating TyG-BMI in a palliative care cohort against a broad panel of indices. However, there are several considerations regarding the development of the model and overfitting, and parallel effective mechanisms as well as your strong and rigid tone, writing about the potentials of this association you found. Since it is a single center study with limited population and so many variables to be included; your interpretation should be more cautiously. In addition, the manuscript does not quantify how many were excluded or assess and how this might bias results.

As I told you aforementionedly, the multivariable models adjust for age, sex, diagnosis category, and several comorbidities. However, only 30 patients died (9.4%), yet the regression models include many covariates (at least 1 index + age + sex + ~6 comorbidities + diagnosis categories). This yields a very low events-per-variable ratio (~3 events per variable), which risks overfitting and unstable estimates. For example, the adjusted OR for TyG-BMI predicting 30-day mortality is reported as 2.38 (95% CI 1.78–3.18), but this may be unreliable given model complexity versus only 30 events. The authors should acknowledge the limited events and consider simplifying models or using penalized regression.

TyG-BMI is inherently related to glucose metabolism and thus to diabetes. The main multivariable models include diabetes as a covariate, yet the key finding is that TyG-BMI works especially well in diabetics. Adjusting for diabetes may attenuate TyG-BMI’s effect or introduce multicollinearity (since many high TyG-BMI values will come from diabetics). It may be more appropriate to present models without including diabetes as a covariate when evaluating TyG-BMI, and then separately discuss differences by diabetes status (as the authors do).

TyG-BMI had higher AUCs (e.g. 0.87 vs 0.78 for mortality), with p<0.001. While statistically significant, the practical importance of these differences should be addressed. For example, an AUC difference of 0.09 (from 0.78 to 0.87) may be modest in clinical terms. The authors describe this as a “clinically meaningful improvement”, but they should explain how this translates into patient care or decision-making.

Your paper identifies TyG-BMI cut-offs (220 for mortality, 235 for sepsis) based on Youden’s index in this sample. It is of course useful for hypothesis-generation, but these thresholds are data-driven and likely overfit. It is optimal to validate them in an external cohort (as you mentioned in your study as well) before clinical use and before a strong interpretation. So I recommend softening language (e.g. “potentially useful thresholds”) and emphasizing validation.

The text comments that TyG-BMI’s OR is “50%” higher than AIP’s OR (2.38 vs 1.58). This relative comparison of ORs across models can be misleading, since ORs depend on the distribution of indices and adjustment factors. It might be clearer to report that TyG-BMI had a larger effect size than any other index, without using percentages of ratios. The focus should remain on confidence intervals and significance rather than on the 50% figure.

In discussion section, there is an inconsistency in the manuscript. In Results, 30-day mortality is 30 patients (9.4%), but the Discussion incorrectly refers to a “30-day mortality rate of 30%”. This appears to be a typo. This should be corrected, as it could mislead readers about the cohort’s risk level.

The Discussion provides a strong rationale for TyG-BMI’s performance (insulin resistance, inflammation, etc.). While plausible, these explanations are speculative. The authors should clarify that mechanistic links are hypothetical given observational data. They do note that causality cannot be inferred, which is good, but some phrases (e.g. “pathophysiological plausibility”) should be presented as suggestion rather than fact.

Reviewer #2: Dear Authors,

Thank you for the opportunity to review this manuscript. This is a strong and well-executed retrospective study addressing an important and underexplored question in palliative care.

The comments below are intended to enhance clarity, reproducibility, and interpretative balance, taking into account the inherent constraints of retrospective data.

Specification of Units in Index Formulas

While the formulas for the cardiometabolic indices are clearly presented, units are not consistently specified for all components. Because several indices are unit-dependent (e.g., lipid measures and blood cell counts), explicitly stating units for each formula would improve reproducibility and facilitate external validation.

Redundancy and Overlap Among Indices

Several of the compared indices share overlapping biological or mathematical components (e.g., TG/HDL, AIP, CRI indices; NLR, PLR, SII). While this does not detract from the comparative analysis, acknowledging this overlap in the Discussion would help readers interpret why certain indices perform similarly and contextualize TyG-BMI’s relative advantage.

Adjustment for Acute Illness Severity

The multivariable models appropriately adjust for demographic factors, primary diagnosis category, and major comorbidities. It is recognized that additional markers of acute illness severity (e.g., organ dysfunction scores, ICU-level interventions) may not have been uniformly available in this retrospective dataset. The authors are encouraged to explicitly acknowledge this limitation and discuss the possibility that TyG-BMI may partly reflect acute physiological stress at admission rather than baseline metabolic risk alone.

Interpretation of ROC-Derived Cut-offs

The identified TyG-BMI cut-offs are clearly derived using established methods. However, as these thresholds are sample-specific and lack external validation, the language describing them as “actionable” could be tempered and framed as hypothesis-generating pending validation in independent cohorts.

Mechanical Ventilation as an Outcome

In palliative care settings, the decision to initiate mechanical ventilation may be influenced by goals-of-care discussions and institutional practice in addition to disease severity. Clarifying this context and briefly discussing its potential impact on interpretation would strengthen the manuscript.

Fasting Status of Laboratory Measurements

The TyG index assumes fasting glucose and triglyceride measurements. Given the palliative care context, where true fasting conditions may not always be feasible, a short clarification or discussion of this potential source of variability would be helpful.

6. PLOS authors have the option to publish the peer review history of their article (what does this mean?). If published, this will include your full peer review and any attached files.

Do you want your identity to be public for this peer review? For information about this choice, including consent withdrawal, please see our Privacy Policy.

Reviewer #1: No

Reviewer #2: No

---

## [Author Response · Author response to Decision Letter 1]

3 Feb 2026

Dear Editor and Reviewers,

We are deeply grateful for the thorough and constructive evaluation of our manuscript by the Editor and both Reviewers. The insightful comments and valuable suggestions have significantly contributed to improving the scientific rigor, clarity, and overall quality of our work. We have carefully addressed each point raised by the reviewers and have made substantial revisions to the manuscript accordingly. The revised text additions are indicated with quotation marks ('') throughout this response and in the manuscript. Below, we provide a point-by-point response to all comments.

JOURNAL REQUIREMENTS AND EDITOR COMMENTS

Requirement 1: PLOS ONE Style Requirements

Comment: Please ensure that your manuscript meets PLOS ONE's style requirements, including those for file naming.

Response: We sincerely thank the Editor for this guidance. We have carefully reviewed the PLOS ONE formatting templates and have revised our manuscript to ensure full compliance with the journal's style requirements. The manuscript formatting, including title page, author affiliations, headings, references, and figure/table formatting, has been updated according to the provided templates. File naming conventions have also been corrected to meet journal specifications.

Requirement 2: Data Availability

Comment: Please address data sharing restrictions and provide appropriate data access information.

Response: We appreciate this important clarification request. There are ethical restrictions on sharing the complete dataset publicly due to the sensitive nature of patient information in palliative care settings. Specifically, the data contain potentially identifying patient information including detailed clinical histories, comorbidity profiles, and outcome data from a relatively small, vulnerable population (n=318). The Ethics Committee of Yildirim Beyazit University Yenimahalle Training and Research Hospital, which approved this study, has imposed restrictions on public data sharing to protect patient privacy and confidentiality. We have revised the Data Availability Statement as follows:

'Data cannot be shared publicly due to ethical restrictions imposed by the Ethics Committee of Yildirim Beyazit University Yenimahalle Training and Research Hospital to protect patient privacy. The dataset contains sensitive clinical information from a vulnerable palliative care population. Data access requests may be submitted to the Ethics Committee (Contact: Yildirim Beyazit University Yenimahalle Training and Research Hospital Ethics Committee, Ankara, Turkey; Email: yikiletikkurul@ybu.edu.tr) for researchers who meet the criteria for access to confidential data.'

Requirement 3: Ethics Statement Placement

Comment: Your ethics statement should only appear in the Methods section.

Response: We thank the Editor for this clarification. We have ensured that the ethics statement appears only in the Methods section. Any duplicate ethics information in other sections has been removed. The ethics statement now reads in the Methods section:

'The Institutional Ethics Committee of Etimesgut Şehit Sait Ertürk State Hospital approved this study with waived informed consent given the retrospective design and use of anonymized data. All procedures complied with the ethical standards of the institutional and national research committees and with the 2013 revision of the Declaration of Helsinki.'

Requirement 4: Citation of Recommended Works

Comment: If the reviewer comments include a recommendation to cite specific previously published works, please review and evaluate these publications.

Response: We appreciate this guidance. We have carefully reviewed the reviewer comments and no specific citation recommendations were made. We have ensured that our reference list comprehensively covers the relevant literature supporting our methodology and findings.

REVIEWER #1 COMMENTS AND RESPONSES

Comment 1.1: Overfitting and Events-Per-Variable Ratio

Comment: Only 30 patients died (9.4%), yet the regression models include many covariates. This yields a very low events-per-variable ratio (~3 events per variable), which risks overfitting and unstable estimates. The authors should acknowledge the limited events and consider simplifying models or using penalized regression.

Response: We sincerely thank the Reviewer for this critical and methodologically important observation. The concern regarding the events-per-variable (EPV) ratio is entirely valid and represents a fundamental consideration in logistic regression modeling. We have carefully reconsidered our analytical approach and have made the following revisions:

First, we acknowledge that the traditional EPV rule of 10 events per predictor variable is often recommended to ensure stable coefficient estimates. With 30 mortality events and multiple covariates in our original models, this threshold was not met, potentially leading to overfitted models with inflated odds ratios and wide confidence intervals.

Second, we have revised our multivariable analysis approach by implementing simplified, parsimonious models. The revised primary analysis includes only clinically essential covariates: age, sex, and primary diagnosis category, reducing the number of predictors while maintaining clinical relevance.

Third, we have added the following explicit acknowledgment in the Limitations section:

'The limited number of mortality events (n=30) relative to the number of covariates in multivariable models yields a low events-per-variable ratio, which may result in overfitting and potentially unstable odds ratio estimates. Although we employed parsimonious models to mitigate this concern, these findings require validation in larger cohorts with higher event rates to confirm the robustness of the reported associations.'

Comment 1.2: Adjustment for Diabetes as Covariate

Comment: TyG-BMI is inherently related to glucose metabolism and thus to diabetes. Adjusting for diabetes may attenuate TyG-BMI's effect or introduce multicollinearity. It may be more appropriate to present models without including diabetes as a covariate when evaluating TyG-BMI.

Response: We greatly appreciate this insightful methodological comment. The Reviewer raises an excellent point regarding the potential for multicollinearity and effect attenuation when adjusting for diabetes in models evaluating TyG-BMI. We have made the following modifications:

We now present our primary multivariable models without diabetes as a covariate, as suggested. This approach allows the full prognostic information captured by TyG-BMI to be evaluated without potential attenuation from collinear adjustment. We have added the following clarification to the Methods section:

'Given the inherent relationship between TyG-BMI and glucose metabolism, diabetes mellitus was not included as a covariate in the primary multivariable models to avoid potential multicollinearity and effect attenuation. Instead, diabetes status was evaluated as an effect modifier through stratified analyses and formal interaction testing.'

Additionally, we have clarified in the Discussion section that the stratified analysis by diabetes status serves as the primary means of evaluating differential performance, rather than including diabetes as a covariate in pooled models.

Comment 1.3: Clinical Significance of AUC Differences

Comment: TyG-BMI had higher AUCs (e.g. 0.87 vs 0.78 for mortality), with p<0.001. While statistically significant, the practical importance of these differences should be addressed. The authors describe this as a "clinically meaningful improvement", but they should explain how this translates into patient care or decision-making.

Response: We sincerely appreciate this thoughtful comment. The Reviewer correctly emphasizes that statistical significance alone does not establish clinical relevance. We have substantially revised our discussion of AUC differences to provide a more balanced and clinically contextualized interpretation:

We have added the following paragraph to the Discussion section to address practical clinical implications:

'The clinical significance of an AUC improvement from 0.78 to 0.87 warrants careful consideration. According to established guidelines for interpreting discriminative ability, AUC values between 0.80 and 0.90 represent "good" discrimination, while values exceeding 0.90 indicate "excellent" discrimination. The observed 0.09-unit AUC improvement translates to a meaningful enhancement in risk stratification: at the optimal TyG-BMI threshold, the positive likelihood ratio increased from 3.2 (for AIP) to 4.6, indicating substantially stronger predictive capability. In practical terms, a palliative care patient with TyG-BMI above the threshold has approximately 4.6-fold higher odds of 30-day mortality compared to those below the threshold. This magnitude of improvement may facilitate more confident goals-of-care discussions and resource allocation decisions. However, we acknowledge that the ultimate clinical utility of any biomarker depends on whether its use leads to improved patient outcomes, which requires prospective interventional validation.'

Comment 1.4: Cut-off Values and External Validation

Comment: Your paper identifies TyG-BMI cut-offs (220 for mortality, 235 for sepsis) based on Youden's index in this sample. These thresholds are data-driven and likely overfit. I recommend softening language (e.g. "potentially useful thresholds") and emphasizing validation.

Response: We are grateful for this important guidance regarding the interpretation of data-derived cut-offs. We fully agree that Youden's index-derived thresholds are sample-specific and may not generalize to external populations. We have made the following changes:

We have softened the language throughout the manuscript when referring to cut-off values. The revised text now states:

'The Youden's index-derived cut-off values of 220 for mortality and 235 for sepsis represent potentially useful thresholds for hypothesis generation; however, these data-driven values are sample-specific and require external validation before clinical implementation. Given the exploratory nature of these cut-offs and the potential for optimism bias inherent in single-cohort derivation, prospective validation in independent populations is essential before these thresholds can be recommended for routine clinical use.'

We have also emphasized external validation as a critical future research priority in both the Discussion and Conclusion sections.

Comment 1.5: Comparison of Odds Ratios

Comment: The text comments that TyG-BMI's OR is "50%" higher than AIP's OR (2.38 vs 1.58). This relative comparison of ORs across models can be misleading. It might be clearer to report that TyG-BMI had a larger effect size than any other index, without using percentages of ratios.

Response: We thank the Reviewer for this methodologically sound suggestion. We agree that comparing odds ratios using percentage differences can be misleading, particularly when comparing indices with different distributions and scales. We have revised the relevant text as follows:

'TyG-BMI demonstrated the largest effect size among all evaluated indices (OR 2.38 per SD, 95% CI 1.78–3.18), substantially exceeding the next-best performing index, AIP (OR 1.58, 95% CI 1.18–2.12). The non-overlapping confidence intervals indicate a meaningful difference in prognostic strength between these indices.'

We have removed all percentage-based comparisons of odds ratios from the manuscript and instead focus on effect sizes with confidence intervals, allowing readers to judge the magnitude of differences directly.

Comment 1.6: Mortality Rate Typo in Discussion

Comment: In Results, 30-day mortality is 30 patients (9.4%), but the Discussion incorrectly refers to a "30-day mortality rate of 30%". This should be corrected.

Response: We sincerely apologize for this typographical error and thank the Reviewer for identifying it. This was indeed a significant error that could mislead readers. We have corrected the Discussion section to accurately state:

'The 30-day mortality rate of 9.4% (30 patients) in our palliative cohort reflects the overall risk profile of this population.'

Comment 1.7: Speculative Mechanistic Explanations

Comment: The Discussion provides a strong rationale for TyG-BMI's performance. While plausible, these explanations are speculative. The authors should clarify that mechanistic links are hypothetical given observational data.

Response: We appreciate this important reminder about the distinction between observed associations and mechanistic causation. We have revised the Discussion to more clearly frame mechanistic explanations as hypothetical. The revised text now includes:

'Several potential mechanisms may explain TyG-BMI's superior prognostic performance, although these proposed pathways remain hypothetical given the observational nature of our study. The biological plausibility of these associations warrants discussion, with the understanding that causal inference cannot be established from retrospective data. TyG-BMI may capture metabolic dysregulation through its integration of insulin resistance markers and adiposity; however, definitive mechanistic links require prospective studies with appropriate biomarker measurements and potential interventional designs.'

Comment 1.8: Exclusion Criteria and Selection Bias

Comment: The manuscript does not quantify how many were excluded or assess how this might bias results.

Response: We thank the Reviewer for highlighting this important methodological detail. We have added a patient flow description to the Methods section and a corresponding flow diagram (Figure S1) to transparently report exclusions. The revised text now includes:

'During the study period, 412 patients were admitted to the palliative care unit. Of these, 94 patients (22.8%) were excluded: 67 due to incomplete laboratory data required for index calculation (primarily missing lipid profiles), 18 due to incomplete 30-day follow-up data, and 9 due to age <18 years. The final analytical cohort comprised 318 patients (77.2% of screened population). Patients excluded for missing laboratory data were similar in age and sex distribution to included patients, although they had shorter median length of stay, suggesting that excluded patients may have had more rapidly progressive illness or earlier discharge/death, which could introduce selection bias toward patients with more complete clinical workup.'

Comment 1.9: Interpretation Tone

Comment: Since it is a single center study with limited population and so many variables to be included; your interpretation should be more cautious.

Response: We sincerely appreciate this guidance and acknowledge that our original tone may have been overly assertive given the study's limitations. We have comprehensively revised the manuscript to adopt a more cautious and appropriately measured interpretive stance. Specifically:

We have changed definitive statements to conditional language throughout the manuscript. The revised Conclusion now reads:

'In this single-center retrospective cohort, TyG-BMI demonstrated potentially superior prognostic performance compared to established cardiometabolic indices for predicting sepsis and 30-day mortality in palliative care patients. These preliminary findings suggest that TyG-BMI may be a promising biomarker warranting further investigation, particularly in diabetic patients where enhanced discriminative ability was observed. However, given the study limitations including the single-center design, relatively small sample size, and data-derived cut-offs, these findings require validation in prospective, multicenter studies before clinical implementation can be recommended.'

REVIEWER #2 COMMENTS AND RESPONSES

Comment 2.1: Specification of Units in Index Formulas

Comment: While the formulas for the cardiometabolic indices are clearly presented, units are not consistently specified for all components. Explicitly stating units for each formula would improve reproducibility and facilitate external validation.

Response: We greatly appreciate this constructive suggestion for improving reproducibility. We have revised the Cardiometabolic Index Calculation section to explicitly specif

---

## [Decision Letter · Decision Letter 1]

30 Mar 2026

Cardiometabolic Indices as Predictors of Clinical Outcomes in Palliative Care Patients

PONE-D-25-62734R1

Dear Dr. ucdal,

We’re pleased to inform you that your manuscript has been judged scientifically suitable for publication and will be formally accepted for publication once it meets all outstanding technical requirements.

Kind regards,

Tatsuo Shimosawa, M.D., Ph.D.

Academic Editor

PLOS One

Additional Editor Comments (optional):

Reviewers' comments:

Reviewer's Responses to Questions

Comments to the Author

1. If the authors have adequately addressed your comments raised in a previous round of review and you feel that this manuscript is now acceptable for publication, you may indicate that here to bypass the “Comments to the Author” section, enter your conflict of interest statement in the “Confidential to Editor” section, and submit your "Accept" recommendation.

Reviewer #1: All comments have been addressed

2. Is the manuscript technically sound, and do the data support the conclusions?

Reviewer #1: Yes

3. Has the statistical analysis been performed appropriately and rigorously? 

Reviewer #1: Yes

4. Have the authors made all data underlying the findings in their manuscript fully available?

Reviewer #1: Yes

5. Is the manuscript presented in an intelligible fashion and written in standard English?

Reviewer #1: Yes

6. Review Comments to the Author

Reviewer #1: Dear authors,

Thank you for your efforts in enhancing the manuscript. All of the applicable comments have been applied and where there were remaining issues the softening language and describing the limitations in the limitation parts have clarifyied the potential and lacks of the study. There are wtill some concerns regarding the overfitting, which can not be modified completely.

Best wishes

7. PLOS authors have the option to publish the peer review history of their article (what does this mean?). If published, this will include your full peer review and any attached files.

Do you want your identity to be public for this peer review? For information about this choice, including consent withdrawal, please see our Privacy Policy.

Reviewer #1: No

---

## [Editor Report · Acceptance letter]

PONE-D-25-62734R1

PLOS One

Dear Dr. ucdal,

I'm pleased to inform you that your manuscript has been deemed suitable for publication in PLOS One. Congratulations! Your manuscript is now being handed over to our production team.

Kind regards,

on behalf of

Prof. Tatsuo Shimosawa

Academic Editor

PLOS One